# Differences between Kidney Transplant Recipients from Deceased Donors with Diabetes Mellitus as Identified by Machine Learning Consensus Clustering

**DOI:** 10.3390/jpm13071094

**Published:** 2023-07-03

**Authors:** Charat Thongprayoon, Jing Miao, Caroline C. Jadlowiec, Shennen A. Mao, Michael A. Mao, Napat Leeaphorn, Wisit Kaewput, Pattharawin Pattharanitima, Supawit Tangpanithandee, Pajaree Krisanapan, Pitchaphon Nissaisorakarn, Matthew Cooper, Wisit Cheungpasitporn

**Affiliations:** 1Division of Nephrology and Hypertension, Department of Medicine, Mayo Clinic, Rochester, MN 55905, USA; charat.thongprayoon@gmail.com (C.T.); miao.jing@mayo.edu (J.M.); supawit_d@hotmail.com (S.T.); pajaree_fai@hotmail.com (P.K.); 2Division of Transplant Surgery, Mayo Clinic, Phoenix, AZ 85054, USA; jadlowiec.caroline@mayo.edu; 3Division of Transplant Surgery, Mayo Clinic, Jacksonville, FL 32224, USA; mao.shennen@mayo.edu; 4Division of Nephrology and Hypertension, Department of Medicine, Mayo Clinic, Jacksonville, FL 32224, USA; mao.michael@mayo.edu (M.A.M.); napat.leeaphorn@gmail.com (N.L.); 5Department of Military and Community Medicine, Phramongkutklao College of Medicine, Bangkok 10400, Thailand; wisitnephro@gmail.com; 6Department of Internal Medicine, Faculty of Medicine, Thammasat University, Pathum Thani 12120, Thailand; pattharawin@hotmail.com; 7Department of Medicine, Division of Nephrology, Massachusetts General Hospital, Harvard Medical School, Boston, MA 02114, USA; pitch.nissa@gmail.com; 8Medical College of Wisconsin, Milwaukee, WI 53226, USA; macooper@mcw.edu

**Keywords:** clustering, diabetic donors, kidney transplant, kidney transplantation, transplantation

## Abstract

Clinical outcomes of deceased donor kidney transplants coming from diabetic donors currently remain inconsistent, possibly due to high heterogeneities in this population. Our study aimed to cluster recipients of diabetic deceased donor kidney transplants using an unsupervised machine learning approach in order to identify subgroups with high risk of inferior outcomes and potential variables associated with these outcomes. Consensus cluster analysis was performed based on recipient-, donor-, and transplant-related characteristics in 7876 recipients of diabetic deceased donor kidney transplants from 2010 to 2019 in the OPTN/UNOS database. We determined the important characteristics of each assigned cluster and compared the post-transplant outcomes between the clusters. Consensus cluster analysis identified three clinically distinct clusters. Recipients in cluster 1 (*n* = 2903) were characterized by oldest age (64 ± 8 years), highest rate of comorbid diabetes mellitus (55%). They were more likely to receive kidney allografts from donors that were older (58 ± 6.3 years), had hypertension (89%), met expanded criteria donor (ECD) status (78%), had a high rate of cerebrovascular death (63%), and carried a high kidney donor profile index (KDPI). Recipients in cluster 2 (*n* = 687) were younger (49 ± 13 years) and all were re-transplant patients with higher panel reactive antibodies (PRA) (88 [IQR 46, 98]) who received kidneys from younger (44 ± 11 years), non-ECD deceased donors (88%) with low numbers of HLA mismatch (4 [IQR 2, 5]). The cluster 3 cohort was characterized by first-time kidney transplant recipients (100%) who received kidney allografts from younger (42 ± 11 years), non-ECD deceased donors (98%). Compared to cluster 3, cluster 1 had higher incidence of primary non-function, delayed graft function, patient death and death-censored graft failure, whereas cluster 2 had higher incidence of delayed graft function and death-censored graft failure but comparable primary non-function and patient death. An unsupervised machine learning approach characterized diabetic donor kidney transplant patients into three clinically distinct clusters with differing outcomes. Our data highlight opportunities to improve utilization of high KDPI kidneys coming from diabetic donors in recipients with survival-limiting comorbidities such as those observed in cluster 1.

## 1. Introduction

End-stage kidney disease (ESKD) is a common and highly morbid disorder that affects both life quality and survival. Kidney transplantation is the best option for most ESKD patients and is superior to other renal replacement therapies such as hemodialysis and peritoneal dialysis in terms of long-term mortality risk [1,2]. In the United States, the number of kidney transplants has increased each year since 2015; however, only a quarter of patients on waitlists received a deceased donor kidney within 5 years due to the ongoing severe shortage of available kidney grafts [3]. Within this context, the transplant community has continued to seek opportunities to increase the size of the available donor pool so as to improve opportunities for transplant. Kidney allografts coming from donors with a higher kidney donor profile index (KDPI) or meeting expanded criteria donor (ECD) status are often imperfect in quality; however, they provide survival and quality of life benefit to patients awaiting transplantation [4]. Utilization of kidneys from donors with diabetes mellitus (DM) has increased remarkably in the last two decades [5]. In the United States, individuals with DM are generally considered ineligible to donate a kidney as living donors due to concern for long-term kidney health. Some studies have demonstrated that deceased donor kidney allografts coming from diabetic donors are at increased risk for delayed graft function, rejection, and inferior graft quality [5,6,7,8]. Despite these risk variables, studies have shown that transplantation with a diabetic donor kidney offers greater survival benefit compared with patients who remained on the waitlist [6].

Although transplant centers remain cautious when using kidney allografts from diabetic and high KDPI donors, there are many advantages in expanding utilization. One advantage is an increased availability of donor organs, which can be critical during the current shortage of donor organs. Utilizing diabetic donor kidneys may also reduce wait times for transplantation because there may be fewer candidates on the waiting list willing to accept such kidneys. Transplant professionals currently determine the utilization of kidneys from donors with DM on a case-by-case basis, considering factors such as the recipient’s kidney disease severity, the availability of appropriate organs, and the quality of the donor organ. The inconsistent results are most likely caused by high heterogenous clinical presentations of both recipients and donors. As such, subgroup analysis of diabetic donor kidney transplant recipients may help identify the specific population and guide selection of candidates suitable for transplantation.

Unsupervised machine learning (ML) is a type of artificial intelligence that enables the analysis of intricate data sets and recognition of patterns without explicit guidance or labeling [7,8]. In the case of kidney transplant recipients, unsupervised ML can be a useful tool for healthcare professionals to identify complex and subtle relationships between patient characteristics, donor characteristics, medical history, treatment history, and outcomes that may not have been apparent using traditional statistical methods [9,10]. The potential applications of unsupervised ML in kidney transplantation include identifying subpopulations of kidney transplant recipients that may benefit from specific interventions. For example, unsupervised ML algorithms can identify groups of kidney transplant recipients with similar clinical characteristics and medical histories but different kidney transplant outcomes [10]. This enables healthcare professionals to identify subpopulations of patients who may benefit from specific treatments or interventions such as adjusting immunosuppressive medication regimens or modifying lifestyle factors. ML consensus clustering is a technique used in data clustering that involves combining the results of multiple clustering algorithms to improve the quality and robustness of clustering results [11,12]. This can help healthcare professionals develop more targeted and effective strategies for managing kidney transplant recipients, leading to better patient outcomes. The utilization of a ML consensus clustering approach could potentially offer healthcare professionals a new perspective on the different phenotypes of kidney transplant recipients who received a kidney from a diabetic donor and have varying outcomes.

The objective of this study was to utilize an unsupervised ML consensus clustering approach to cluster kidney transplant patients who received kidneys from deceased diabetic donors based on a wide range of recipient-, donor-, and transplant-related variables, and subsequently examine the post-transplant outcomes among the distinct clusters identified through the ML algorithm.

## 2. Materials and Methods

### 2.1. Data Source and Study Population

The study population consisted of adult kidney transplant recipients in the United States between 2010 and 2019, identified through the Organ Procurement and Transplantation Network (OPTN)/United Network for Organ Sharing (UNOS) database. The inclusion criteria encompassed kidney transplants from deceased donors with diabetes. The database did not specify the type of diabetes in donors; therefore, any diabetic donors, regardless of diabetic type, were included. Approval for this study was obtained from the Mayo Clinic Institutional Review Board (IRB number 21-007698).

### 2.2. Data Collection

A comprehensive set of recipient-, donor-, and transplant-related characteristics, as outlined in Table 1, were extracted from the OPTN/UNOS database for subsequent incorporation into the clustering analysis. All variables exhibited missing data of less than 10%. To address the missing values, the multiple imputation by chained equation (MICE) method [13] was employed for imputing the missing data.

### 2.3. Clustering Analysis

In this study, an unsupervised machine learning (ML) technique was employed to categorize the clinical phenotypes of deceased diabetic donor kidney transplant patients [14]. To ensure meaningful clinical clusters, a consensus clustering approach was utilized. The clustering process involved applying a pre-specified subsampling parameter of 80% with 100 iterations, and considering a range of potential clusters (k) from 2 to 10. The objective was to avoid generating an excessive number of clusters that lacked clinical significance. The determination of the optimal number of clusters was based on several factors, including the examination of the consensus matrix (CM) heat map, the cumulative distribution function (CDF), cluster-consensus plots incorporating within-cluster consensus scores, and the proportion of ambiguously clustered pairs (PAC). The within-cluster consensus score, ranging from 0 to 1, was calculated as the average consensus value among pairs of individuals belonging to the same cluster [12]. A higher score indicated greater cluster stability. In contrast, the PAC, also ranging from 0 to 1, represented the proportion of sample pairs with consensus values falling within predetermined boundaries [15]. A lower PAC value indicated better cluster stability [15]. For further details regarding the consensus cluster algorithms employed in this study for reproducibility purposes, please refer to the Appendix A.

### 2.4. Outcomes

The study evaluated various outcomes following kidney transplantation from diabetic donors, including primary non-function, delayed graft function, acute rejection within 1 year, patient death, and death-censored graft failure within 1 and 5 years post-transplant. Death-censored graft failure was defined as the requirement for dialysis or re-transplantation, while considering patients censored for death or at the most recent follow-up date available in the OPTN/UNOS database.

### 2.5. Statistical Analysis

Following the assignment of each kidney transplant recipient from diabetic donors into clusters through consensus clustering analysis, a comparative analysis of clinical characteristics and post-transplant outcomes was conducted among the assigned clusters. The differences in clinical characteristics among the clusters were assessed using the Chi-squared test for categorical variables and analysis of variance (ANOVA) for continuous variables. The identification of key characteristics for each cluster was determined by measuring the standardized mean difference (SMD) between each cluster and the overall cohort, with a predetermined cutoff of >0.3. Rather than relying solely on *p*-values, we chose this threshold based on the effect size considered clinically significant for the variables under examination. By adopting this approach [10], we aimed to capture substantial differences among the clusters while taking into account the magnitude of these differences. Patient survival and death-censored graft survival were estimated using Kaplan–Meier analysis, and comparisons among the clusters were made using the log-rank test. Hazard ratios (HR) for patient death and death-censored graft failure were calculated using Cox proportional hazard analysis, while odds ratios (OR) for primary non-function, delayed graft function, and 1-year rejection were determined using logistic regression analysis. Since the consensus clustering approach deliberately generated clinically distinct clusters, the associations of the assigned clusters with post-transplant outcomes were not adjusted for patient characteristics. All statistical analyses were performed using R, version 4.0.3 (Rstudio, Inc., Boston, MA, USA; http://www.rstudio.com/, accessed on 18 January 2023), including the ConsensusClusterPlus package (version 1.46.0) for consensus clustering analysis and the MICE command in R for multivariable imputation by chained equation [13].

## 3. Results

### 3.1. Clinical Characteristics of Each Cluster of Kidney Transplant Recipients from Diabetic Donors

During the study period from 2010 to 2019, 158,367 adult patients underwent kidney transplants. Of these, 7876 (5%) received kidney transplants from a deceased donor with diabetes. Among these donors, 69% had concurrent history of hypertension, and 31% had a KDPI of ≥85%. Therefore, consensus clustering analysis was performed in 7876 kidney transplant recipients from diabetic deceased donors.

Figure 1A displays the cumulative distribution function (CDF) plot, illustrating the consensus distributions for each cluster of kidney transplant patients from deceased donors with diabetes. The delta area plot (Figure 1B) represents the relative changes in the area under the CDF curve. The most significant alterations in the area occurred between k = 2 and k = 5, after which the relative increase in the area became less pronounced. The CM heat map (Figure 1C, Appendix A) demonstrates the identification of cluster 3 with distinct boundaries by the machine learning algorithm, indicating robust cluster stability across multiple iterations. The mean cluster consensus score was found to be highest in cluster 3 (Figure 2A). Additionally, favorable low partition around medoids clustering (PAC) was observed for the three clusters (Figure 2B). Thus, the consensus clustering analysis identified three clusters that best represented the data pattern of diabetic deceased donor kidney transplant patients.

There were 2903 (37%) patients in cluster 1, 1687 (9%) patients in cluster 2, and 4286 (54%) patients in cluster 3. Table 1 shows recipient-, donor-, and transplant-related characteristics of diabetic donor kidney transplant recipients according to the assigned clusters. According to standardized mean differences, shown in Figure 3, cluster 1 recipients were older in age (mean age 64 years) first-time kidney transplant recipients (99%). Kidney allografts in cluster 1 were more likely to come from older (mean age 45 years) hypertensive (89%) donors who met ECD criteria (78%). Cluster 1 donors were more likely to have a cerebrovascular cause of death (63%) and a high KDPI (77% had KDPI ≥ 85%). Recipients in cluster 1 were more likely to be diabetic compared to cluster 2 and 3 (55% vs. 21% and 38%, respectively). Cluster 2 recipients were younger (mean age 49 years) re-transplants (100%) with higher panel reactive antibodies (PRA) (median 88). Cluster 2 recipients received kidneys from younger (mean age 44 years), non-ECD (12%) donors with low number of HLA mismatch (median 4). Cluster 3 recipients were first-time kidney transplant patients who received kidneys from younger (mean age 42 years) non-ECD (98%) diabetic donors with lower KDPI (3% had KDPI ≥ 85%).

### 3.2. Post-Transplant Outcomes of Each Cluster of Kidney Transplant Recipients from Diabetic Donors

Table 2 presents the outcomes based on the assigned clusters. Notably, cluster 3 exhibited superior outcomes compared to clusters 2 and 3. Cluster 3 demonstrated the highest rates of both overall patient survival (96.5% at 1-year and 81.4% at 5-year, as depicted in Figure 4A) and death-censored graft survival (95.4% at 1-year and 82.1% at 5-year, as shown in Figure 4B). Using Cluster 3 as the reference group, cluster 1 had higher incidence of primary non-function (OR 1.65; 95% CI 1.0–2.62), delayed graft function (OR 1.21–1.10–1.34), 5-year patient death (HR 1.92; 95% CI 1.69–2.17), and death-censored graft failure (HR 1.46; 95% CI 1.28–1.66), whereas Cluster 2 had higher delayed graft function (OR 1.27; 95% CI 1.07–1.50) and death-censored graft failure (HR 1.36; 95% CI 1.09–1.67), but comparable primary non-function and patient death.

## 4. Discussion

We have successfully categorized kidney transplant recipients who received deceased donor kidneys from diabetic donors into 3 clusters with distinct clinical features and outcomes by using an unsupervised ML approach. Compared to clusters 1 and 2, cluster 3 recipients had superior outcomes specific to death-censored graft survival and patient survival at both 1 and 5 years. Compared to the other clusters, cluster 3 recipients had more favorable recipient and donor characteristics. All recipients in cluster 3 were first-time kidney transplant patients who received kidney allografts from younger donors with a lower KDPI score.

Cluster 1 recipients accounted for ~40% of this study cohort. Cluster 1 recipients had the least favorable recipient and donor characteristics. These characteristics included older age and presence of diabetes. Donor-recipient pairing was evident in cluster 1 and recipients in cluster 1 were more likely to receive higher KDPI allografts. Transplant outcomes, including survival, are known to be reduced in older recipients with comorbidities [16,17]. Use of higher KDPI kidney allografts in older recipients highlights opportunities to improve access and equity in kidney transplantation [18]. This compliments OPTN data, which has shown that outcomes are better with transplantation compared with remaining on the waitlist [6]. Use of older, higher KPDI, diabetic deceased donor kidney allografts is likely of greatest benefit to older recipients with minimal qualifying time, lack of available living donors, and presence of survival-limiting comorbidities. Although use of allografts with high KPDI characteristics are unlikely to be advantageous for younger recipients, diabetic kidney allografts from younger donors are likely to be of suitable quality. As suggested by findings from this unsupervised ML, kidney allografts with donor characteristics observed in clusters 2 and 3 should be considered for all waitlist patients, including those who are younger in age. Careful screening of the allograft, including consideration of a procurement biopsy, can help guide clinical decision making on the appropriate recipient. Overall, outcomes in cluster 1 highlight opportunities to increase utilization of higher KDPI kidneys in older recipients with significant survival-limiting comorbidities. Patient survival, independent of graft quality, was the largest factor responsible for influencing outcomes. Use of a higher quality deceased donor is unlikely to have yielded differences in patient survival.

Recipients and donors in cluster 3 had the most favorable characteristics including younger age. The favorable patient and death-censored graft survival observed in cluster 3 highlights the variability observed in diabetic deceased donor kidney allografts. Duration of diabetes and medical management influence allograft quality. The importance of procurement biopsies is better established in high KDPI ECD donors; however, use of procurement biopsies in standard KDPI donors (KDPI < 85%) can also be of value, particularly for donors with diabetes.

Although recipients in cluster 2 were younger, and also had more favorable characteristics, the cluster 2 cohort had decreased death-censored graft survival. All recipients in cluster 2 were re-transplant patients and this likely played a role in the graft survival observed. Similar to cluster 3, cluster 2 donors also had more favorable characteristics, such as younger age and lower KDPI, despite the presence of diabetes. It is likely that factors such as infection and rejection, often associated with re-transplantation, may have been responsible for the reduced graft survival observed.

Overall, the findings from this unsupervised ML approach highlight opportunities to further improve the utilization of deceased donor kidney allografts coming from diabetic donors. Although diabetic donor kidneys are increasingly accepted, our data still shows that only a small percentage (5%) of transplanted kidneys were diabetic donor kidneys, which is similar to other reports ranging from 3.5% to 8.8% [5,19,20,21,22,23]. The data from the UNOS registry suggested that approximately 40–50% of diabetic donor kidneys have been discarded each year from 2008 to 2019, and diabetic ECD donor kidneys had higher discarded rates, approximately 60% [3,24]. The current findings suggest that factors intrinsic to the recipient, including diabetes status and re-transplantation, more significantly impact graft and patient survival. Our study identified three different subpopulations of diabetic donor kidney recipients with distinct features and outcomes, which help us determine the recipients with high risks. These findings prompt us to better understand, evaluate, and allocate diabetic donor kidneys to shorten the waitlist and reduce the risk of death on the waitlist, although the impact of diabetic donor kidneys on patient and graft survival should be further investigated in large, prospective, and randomized studies.

The findings from this ML approach suggest that there are opportunities to further improve the utilization of deceased donor kidney allografts coming from diabetic donors. Factors intrinsic to the recipient, including diabetes status and re-transplantation, rather than presence of diabetes in the donor, appear to play more significant roles in influencing graft and patient survival. Unsupervised ML has the potential to provide healthcare professionals with a novel viewpoint on the distinct phenotypes of kidney transplant recipients who received kidneys from donors with diabetes and experienced disparate outcomes, and thus significantly improved the care and outcomes for these complex patients.

Limited biopsy data and missing reported data from the UNOS are important limitations of this study. Biopsies play a crucial role in determining the quality and suitability of kidneys for transplantation, including the utilization of kidneys from diabetic donors. However, despite their importance, biopsy data are often incomplete or unavailable. The quality of a kidney from a donor with diabetes can be influenced by several factors, such as the duration and severity of the donor’s diabetes, the presence of diabetic complications, and other comorbidities. Kidneys from diabetic donors may have microscopic changes, such as arteriolar hyalinosis and interstitial fibrosis, which can reduce the functional capacity of the kidney. Additionally, donors who had well-managed blood sugar levels and exhibit no signs or symptoms of renal impairment, such as elevated levels of protein in their urine or decreased kidney function, may not suffer from diabetic kidney disease. Our dataset had certain limitations, including the absence of information on coronary artery disease, post-transplant diabetes, and CMV prophylaxis management [25]. These factors are important considerations in the context of kidney transplantation outcomes. Therefore, future research endeavors should focus on incorporating these crucial pieces of information to gain a more comprehensive understanding of the characteristics of both recipients and donors in the UNOS database. By including these data, a more detailed analysis can be conducted to assess their impact on transplant outcomes and potentially enhance patient care in the field of kidney transplantation. Additional studies could expand upon the information that is currently available, allowing for a more comprehensive understanding of recipient and donor-related factors and their impact on transplant outcomes. Additionally, we utilized the SMD technique with a predetermined threshold exceeding 0.3 to identify the distinct characteristics associated with each cluster [26,27,28,29]. Instead of solely relying on *p*-values, our decision to use this cutoff value was informed by the effect size that was deemed clinically significant for the variables being investigated. This methodological approach was implemented to ensure that we captured meaningful differences among the clusters while considering the magnitude of these differences [10]. For instance, working income did not emerge as a distinct clinical characteristic specific to each assigned cluster. However, employment status and its associated factors, such as working income, may indeed play a role in post-transplant outcomes and adherence [30]. Future studies that incorporate data on employment status and post-transplant adherence could provide further insights into the relationship between working income, adherence, and transplant outcomes. Finally, our dataset did not contain waitlisted patients [31]. For instance, it is noteworthy that the impact of type 2 diabetes on the survival of waitlisted individuals is substantial [32]. As a result, it may be necessary to modify allocation policies for type 2 diabetes patients to account for the heightened risk of mortality and the possibility of their waitlist being suspended due to the presence of other medical conditions. Future studies to apply this machine learning approach in waitlisted patients would be of interest to better identify waitlisted patients who would have more survival benefit from receiving kidney transplant from donors with diabetes, high KDPI, or ECD status.

## 5. Conclusions

Our study utilized an unsupervised machine learning approach to cluster recipients of diabetic deceased donor kidney transplants into three clinically distinct groups based on recipient-, donor-, and transplant-related characteristics. Our findings suggest that recipients in Cluster 1, who were characterized by older age, comorbid diabetes mellitus, and high-KDPI kidneys, were associated with poorer post-transplant outcomes. This highlights the need for improved utilization of high-KDPI kidneys coming from diabetic donors in recipients with survival-limiting comorbidities. Additionally, Cluster 2 recipients, who were younger re-transplant patients with higher PRA, were associated with higher incidence of delayed graft function and death-censored graft failure. These results may aid in identifying subgroups of recipients who require closer monitoring and tailored interventions to improve their post-transplant outcomes.

## Figures and Tables

**Figure 1 jpm-13-01094-f001:**
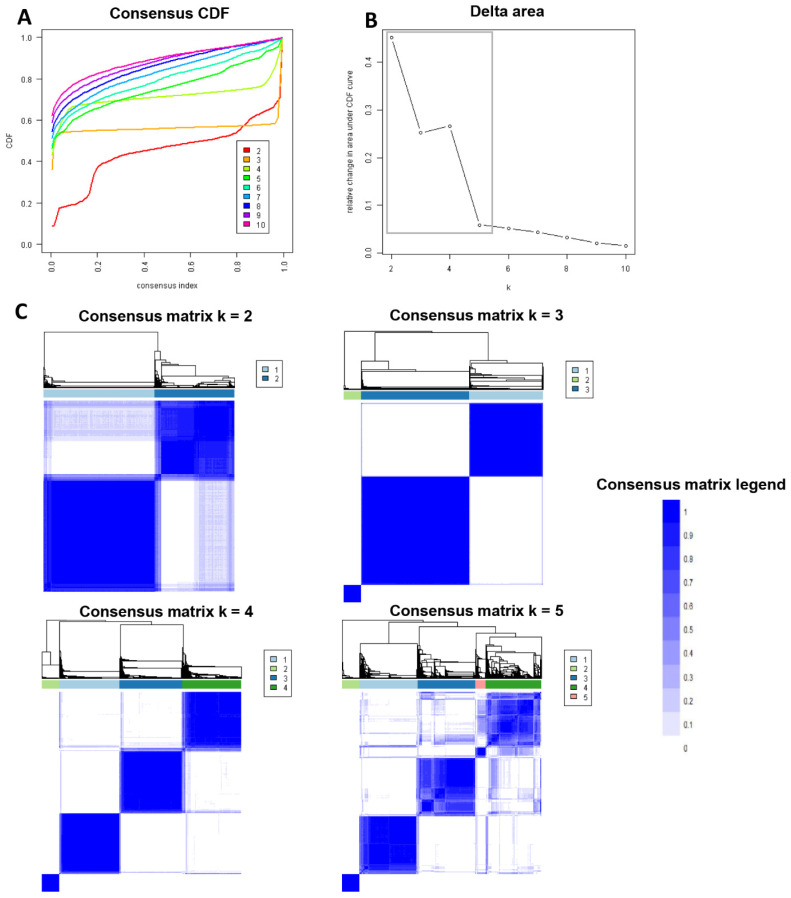
Visual representations of the analysis results. In (**A**), the cumulative distribution function (CDF) plot showcases the consensus distributions for each value of k. The relative changes in the area under the CDF curve are depicted in the delta area plot (**B**). The consensus matrix heat map (**C**) portrays the consensus values of each cluster, displayed on a color scale ranging from white to blue.

**Figure 2 jpm-13-01094-f002:**
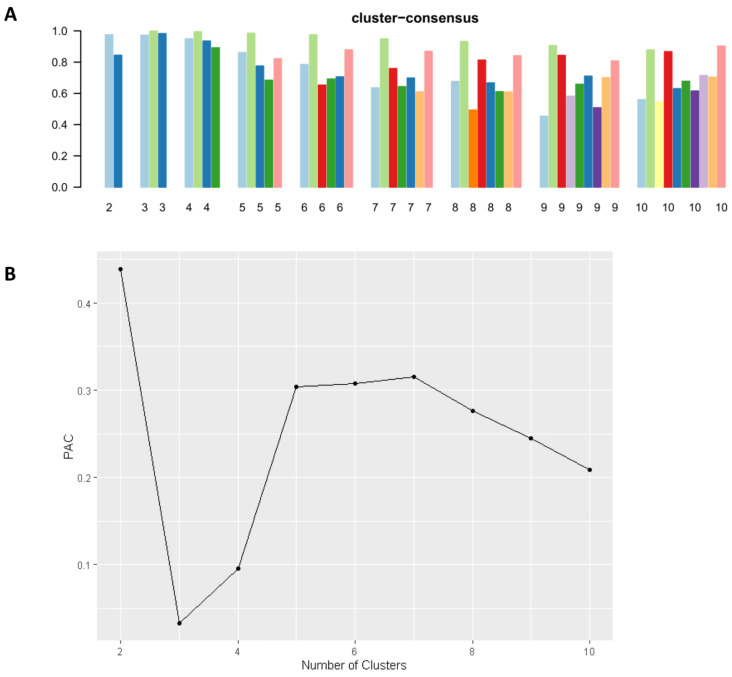
Key findings from the analysis. In (**A**), the bar plot showcases the average consensus score across different numbers of clusters (k ranging from two to ten). Different color columns indicate different cluster groups. Additionally, (**B**) presents the assessment of ambiguously clustered pairs using PAC values.

**Figure 3 jpm-13-01094-f003:**
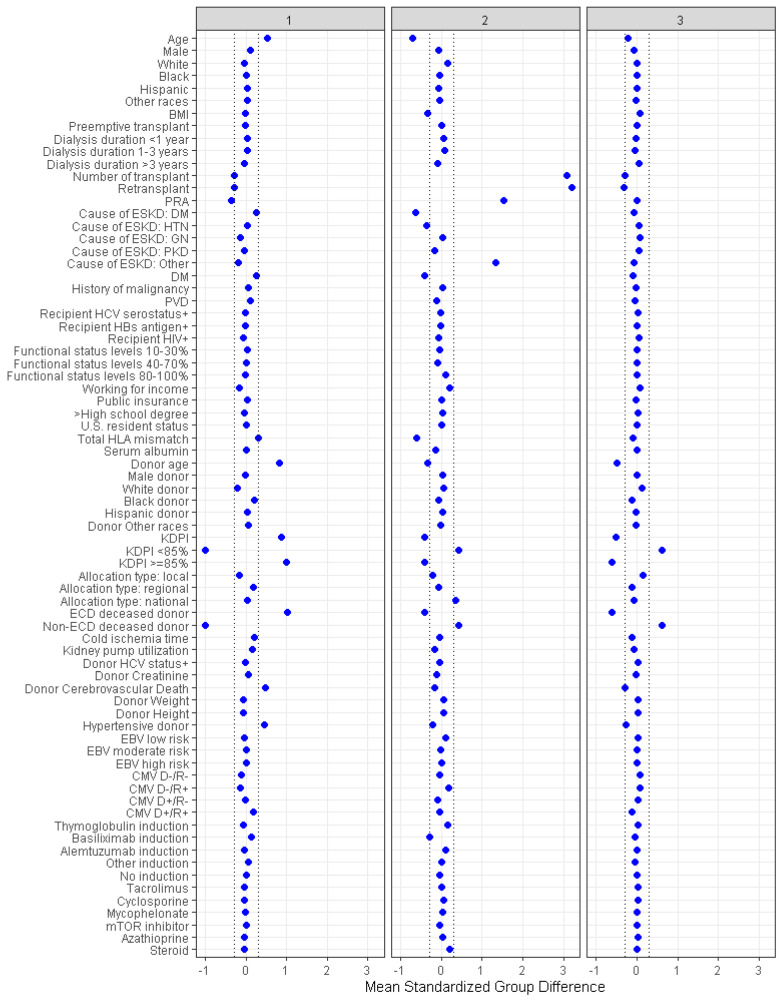
Plot of standardized mean difference, which highlights the clinical characteristics specific to each cluster.

**Figure 4 jpm-13-01094-f004:**
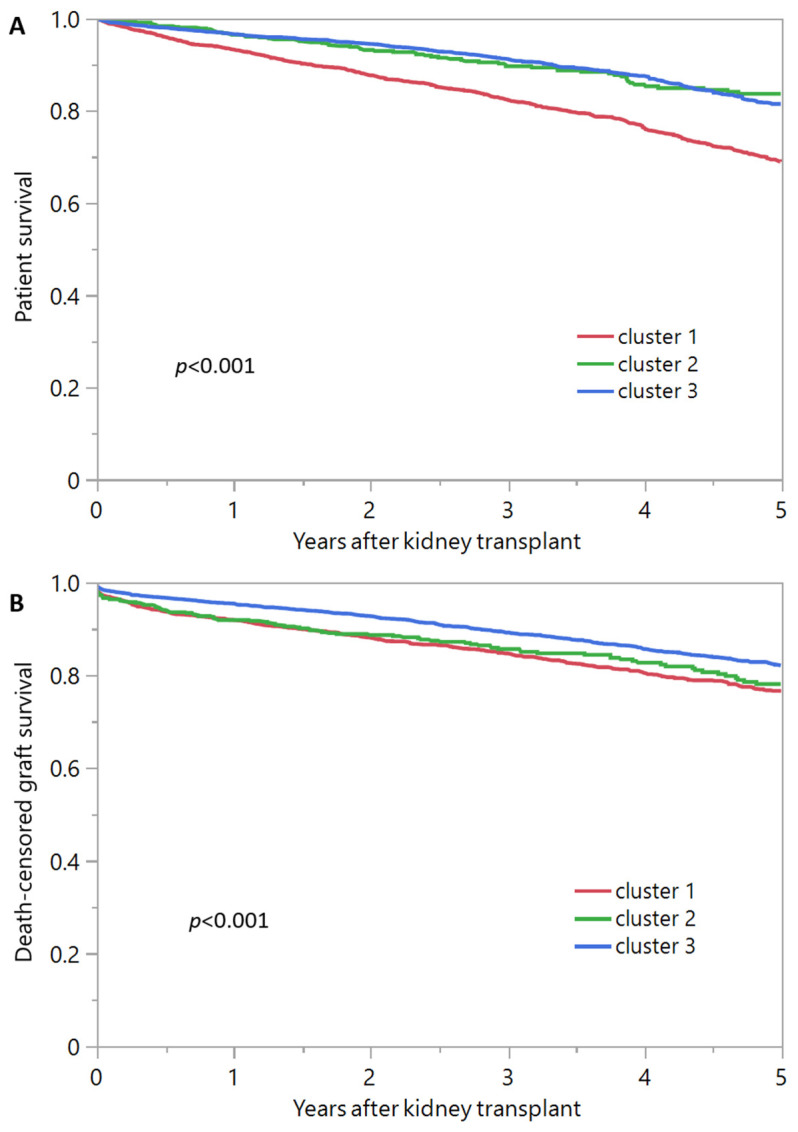
Outcomes of patient survival (**A**) and death-censored graft survival (**B**) specifically for the three distinct clusters of kidney transplant recipients from deceased donors with diabetes.

**Table 1 jpm-13-01094-t001:** Clinical characteristics according to clusters of kidney transplant recipients from diabetic donor.

	All (*n* = 7876)	Cluster 1 (*n* = 2903)	Cluster 2 (*n* = 687)	Cluster 3 (*n* = 4286)	*p*-Value
Recipient Age (year)	58 ± 12	64 ± 8	49 ± 13	55 ± 12	<0.001
Recipient male sex	4955 (63)	1992 (69)	408 (59)	2555 (60)	<0.001
Recipient race - White - Black - Hispanic - Other					0.001
	3187 (40)	1114 (38)	325 (47)	1748 (41)	
	2579 (33)	952 (33)	210 (31)	1417 (33)	
	1395 (18)	547 (19)	100 (15)	748 (17)	
	715 (9)	290 (10)	52 (8)	373 (9)	
Body mass index (kg/m^2^)	28.7 ± 5.2	28.6 ± 4.9	26.9 ± 5.1	29.0 ± 5.4	<0.001
Kidney re-transplant	702 (9)	15 (1)	687 (100)	0 (0)	<0.001
Dialysis duration - Preemptive - <1 year - 1–3 years - >3 years					<0.001
	744 (9)	257 (9)	68 (10)	419 (10)	
	633 (8)	263 (9)	65 (9)	305 (7)	
	1800 (23)	717 (25)	178 (26)	905 (21)	
	4699 (60)	1666 (57)	376 (55)	2657 (62)	
Cause of end-stage kidney disease - Diabetes mellitus - Hypertension - Glomerular disease - PKD - Other					<0.001
	2656 (34)	1319 (45)	27 (4)	1310 (31)	
	2121 (27)	816 (28)	72 (10)	1233 (29)	
	1208 (15)	305 (11)	114 (17)	789 (18)	
	6626 (8)	201 (7)	24 (3)	401 (9)	
	1265 (16)	262 (9)	450 (66)	553 (13)	
Comorbidity - Diabetes mellitus - Malignancy - Peripheral vascular disease					
	3348 (43)	1586 (55)	147 (21)	1615 (38)	<0.001
	718 (9)	306 (11)	66 (10)	346 (8)	0.002
	790 (10)	379 (13)	45 (7)	366 (9)	<0.001
PRA, median (IQR)	0 (0, 24)	0 (0, 0)	88 (46, 98)	0 (0, 25)	<0.001
Positive HCV serostatus	341 (4)	110 (4)	27 (4)	204 (5)	0.12
Positive HBs antigen	147 (2)	51 (2)	11 (2)	85 (2)	0.68
Positive HIV serostatus	69 (1)	8 (0.3)	1 (0.2)	60 (1)	<0.001
Functional status - 10–30% - 40–70% - 80–100%					0.03
	21 (0.3)	11 (0.4)	0 (0)	10 (0.2)	
	3326 (42)	1249 (43)	257 (37)	1820 (42)	
	4529 (58)	1643 (57)	430 (63)	2456 (57)	
Working income	1821 (23)	477 (16)	220 (32)	1124 (26)	<0.001
Public insurance	6272 (80)	2353 (81)	547 (80)	3372 (79)	0.049
US resident	7834 (99)	2890 (100)	684 (100)	4260 (99)	0.62
Undergraduate education or above	3888 (49)	1375 (47)	351 (51)	2162 (50)	0.02
Serum albumin (g/dL)	4.0 ± 0.6	4.0 ± 0.6	3.9 ± 0.6	4.0 ± 0.6	0.001
Kidney donor status - Non-ECD deceased - ECD deceased					<0.001
	5413 (69)	630 (22)	603 (88)	4180 (98)	
	2463 (31)	2273 (78)	84 (12)	106 (2)	
Donor age	48 ± 12	58 ± 6.3	44 ± 11	42 ± 11	<0.001
Donor male sex	4296 (55)	1553 (54)	382 (56)	2361 (55)	0.35
Donor race - White - Black - Hispanic - Other					<0.001
	4846 (62)	1493 (51)	438 (64)	2915 (68)	
	1246 (16)	686 (24)	88 (13)	472 (11)	
	1373 (17)	542 (19)	130 (19)	701 (16)	
	411 (5)	182 (6)	31 (5)	198 (5)	
Donor weight (kg)	94 ± 26	92 ± 23	95 ± 28	95 ± 27	<0.001
Donor height (cm)	170 ± 11	169 ± 11	170 ± 11	170 ± 12	<0.001
Donor hypertension	5414 (69)	2590 (89)	406 (59)	2418 (56)	<0.001
Donor positive HCV serostatus	147 (2)	46 (2)	8 (1)	93 (2)	0.07
Donor cerebrovascular death	3100 (39)	1828 (63)	210 (31)	1062 (25)	<0.001
Donor creatinine (mg/dL)	1.2 ± 0.9	1.3 ± 0.8	1.1 ± 0.7	1.2 ± 0.9	<0.001
KDPI ≥ 85	2415 (31)	2226 (77)	79 (12)	110 (3)	<0.001
HLA mismatch, median (IQR)	4 (4, 5)	5 (4, 5)	4 (2, 5)	4 (3, 5)	<0.001
Cold ischemia time (hours)	19.3 ± 9.2	21.2 ± 9.5	18.9 ± 8.7	18.1 ± 8.8	<0.001
Kidney on pump	4546 (58)	1888 (65)	341 (50)	2317 (54)	<0.001
Allocation type - Local - Regional - National					<0.001
	5374 (68)	1754 (60)	401 (58)	3219 (75)	
	1313 (17)	690 (24)	95 (14)	528 (12)	
	1189 (15)	459 (16)	191 (28)	539 (13)	
EBV status - Low risk - Moderate risk - High risk					0.004
	34 (0.4)	4 (0.1)	8 (1)	22 (1)	
	7199 (91)	2660 (92)	622 (91)	3917 (91)	
	643 (8)	239 (8)	57 (8)	347 (8)	
CMV status - D−/R− - D−/R+ - D+/R+ - D+/R−					<0.001
	865 (11)	224 (8)	67 (10)	574 (13)	
	1850 (23)	508 (18)	214 (31)	1128 (26)	
	3681 (47)	1633 (56)	302 (44)	1746 (41)	
	1480 (19)	538 (19)	104 (15)	838 (20)	
Induction immunosuppression - Thymoglobulin - Alemtuzumab - Basiliximab - Other - No induction					
	4550 (58)	1583 (55)	448 (65)	2519 (59)	<0.001
	1226 (16)	420 (14)	132 (19)	674 (16)	0.008
	1593 (20)	745 (26)	61 (9)	787 (18)	<0.001
	178 (2)	90 (3)	16 (2)	72 (2)	<0.001
	691 (9)	254 (9)	52 (8)	385 (9)	0.48
Maintenance Immunosuppression - Tacrolimus - Cyclosporine - Mycophenolate - Azathioprine - mTOR inhibitors - Steroid					
	7081 (90)	2576 (89)	617 (90)	3888 (91)	0.02
	147 (2)	34 (1)	17 (2)	96 (2)	0.002
	7194 (91)	2634 (91)	634 (92)	3926 (92)	0.29
	23 (0.3)	3 (0.10	3 (0.4)	17 (0.4)	0.06
	96 (1)	36 (1)	4 (1)	56 (1)	0.27
	5328 (68)	1900 (65)	527 (77)	2901 (68)	<0.001

Abbreviations used in the study are defined as follows: BMI (Body mass index), CMV (Cytomegalovirus), CM (centimeter), D (Donor), EBV (Epstein–Barr virus), ECD (Extended criteria donor), HBs (Hepatitis B surface), HCV (Hepatitis C virus), HIV (Human immunodeficiency virus), IQR (interquartile range), KDPI (Kidney donor profile index), KG (kilogram), mTOR (Mammalian target of rapamycin), PKD (Polycystic kidney disease), PRA (Panel reactive antibody), and R (Recipient).

**Table 2 jpm-13-01094-t002:** Post-transplant outcomes according to the clusters.

	Cluster 1	Cluster 2	Cluster 3
Primary non-function	39 (1.3)	4 (0.6)	35 (0.8)
OR for primary non-function	1.65 (1.05–2.62)	0.71 (0.25–2.01)	1 (ref)
Delayed graft function	991 (34)	241 (35)	1282 (30)
OR for delayed graft function	1.21 (1.10–1.34)	1.27 (1.07–1.50)	1 (ref)
1-year survival	93.2%	96.4%	96.5%
HR for 1-year death	2.01 (1.60–2.51)	1.02 (0.65–1.60)	1 (ref)
5-year survival	68.8%	83.6%	81.4%
HR for 5-year death	1.92 (1.69–2.17)	0.97 (0.75–1.24)	1 (ref)
1-year death-censored graft survival	91.9%	91.8%	95.4%
HR for 1-year death-censored graft failure	1.81 (1.48–2.20)	1.80 (1.32–2.45)	1 (ref)
5-year death-censored graft survival	76.6%	78.1%	82.1%
HR for 5-year death-censored graft failure	1.46 (1.28–1.66)	1.36 (1.09–1.67)	1 (ref)
1-year acute rejection	183 (6.3)	53 (7.7)	253 (5.9)
OR for 1-year acute rejection	1.07 (0.88–1.31)	1.33 (0.98–1.81)	1 (ref)

## Data Availability

The data used in this study can be obtained upon reasonable request to the corresponding author.

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
