# Peer review of "Differences between Kidney Transplant Recipients from Deceased Donors with Diabetes Mellitus as Identified by Machine Learning Consensus Clustering"

_jpm, 2023, doi:10.3390/jpm13071094_

Round 1
Reviewer 1 Report
Due to the shortage of organs for transplantation, suboptimal organs are used for this purpose. An example are kidneys from diabetic deceased donors. In this paper, the authors used unsupervised machine learning to analyze characteristics of diabetic deceased donors and recipients of their organs, as well as data on the outcomes. As a result, 3 clusters of patients were identified with distinct clinical features and outcomes. The study was well-designed and performed. The article is well-written and clear. The conclusions are supported by obtained results. Additionally, the results are of clinical importance, as features were identified that should spawn individualized management of renal transplant recipients. In my opinion it is a very good paper; I only suggest a minor corrections:
The meaning of abbreviations used should be explained at their first use (e.g. KDPI, ECD).
Author Response
Due to the shortage of organs for transplantation, suboptimal organs are used for this purpose. An example are kidneys from diabetic deceased donors. In this paper, the authors used unsupervised machine learning to analyze characteristics of diabetic deceased donors and recipients of their organs, as well as data on the outcomes. As a result, 3 clusters of patients were identified with distinct clinical features and outcomes. The study was well-designed and performed. The article is well-written and clear. The conclusions are supported by obtained results. Additionally, the results are of clinical importance, as features were identified that should spawn individualized management of renal transplant recipients. In my opinion it is a very good paper; I only suggest a minor corrections:
The meaning of abbreviations used should be explained at their first use (e.g. KDPI, ECD).
Response: We would like to express our sincere gratitude for reviewing our manuscript and providing us with your critical evaluation. We greatly appreciate your positive feedback and your recognition of the value and clinical importance of our study.
We have taken your suggestion regarding the meaning of abbreviations used in the manuscript into consideration, and we have made the necessary revisions. We now provide explanations for abbreviations such as KDPI (Kidney Donor Profile Index) and ECD (Expanded Criteria Donor) at their first use in the text. This clarification will enhance the readability and understanding of the manuscript for our readers.
Once again, we thank you for your valuable feedback and for acknowledging the rigorous design and execution of our study. We are pleased to hear that you found the article well-written, clear, and supported by the obtained results. We share your belief that the identification of distinct clinical features and outcomes in the three clusters of patients will contribute to individualized management strategies for renal transplant recipients, ultimately improving patient care and outcomes.
Your input has been invaluable, and we have incorporated your suggestions into the revised manuscript. We hope that these revisions further enhance the clarity and impact of our research.
Thank you again for your time and effort in reviewing our manuscript.

Reviewer 2 Report
I read with interest the paper by Thongprayoon et al. about the differences between kidney transplant recipients from deceased donors with diabetes mellitus. The adoption of an unsupervised machine learning approach allows the authors to stratify patient outcomes in three clusters, stressing the importance of improving the utilization of high KDPI kidneys from diabetic donors in some circumstances. Considering that the topic is a critical issue in the transplant community, I suggest assessing additional data to improve the generalizability of the paper.
- Do diabetic donors include subjects only with T2D or also T1D? Please explain this question in the methods.
- Many studies reported the death censored graft survival in still waitlisted patients with similar characteristics to assess the advantage of transplanting a kidney with high KDPI or significant comorbidity (see in example doi: 10.2215/CJN.06100617). If available, these data should be added; alternatively, comment on this point in the limitation of the study.
- Some data should be better explained or expanded (i.e., differences in serum albumin between groups are small [4.0±0.6, 3.9±0.6, 4.0±0.6] but statistically significant; Coronary artery disease has not been assessed as a comorbid condition)
- Working income differs among clusters, may reflect employment status, and could partially influence post-transplant adherence (e.g., doi: 0.23736/S2724-6051.21.04244-2). Please consider this point in the discussion.
- T2D significantly affects survival also on waitlisted patients; accordingly, allocation policies in T2D patients may be adjusted according to increased risk of mortality and waitlist suspension due to comorbidities (see doi: 10.1038/s41598-020-78938-3). Please consider this aspect and the inclusion of diabetic donors for these patients in the discussion.
- CMV serostatus differs between clusters and may influence the risk of CMV viremia, potentially impacting patient survival (see, for example, doi: 10.3390/microorganisms11020458). Please report prophylaxis management and appropriate comments in the discussion.
- Correct typos (e.g., Recipients in cluster 2 (N=687) ware younger (49±13 years) 33 and all were re-transplant patients with higher PRA…; portrays the consensus values of each cluster, displayed on 204 a color scale ranging from white to blue..).
Minor editing of English language required (correct typos).
Author Response
I read with interest the paper by Thongprayoon et al. about the differences between kidney transplant recipients from deceased donors with diabetes mellitus. The adoption of an unsupervised machine learning approach allows the authors to stratify patient outcomes in three clusters, stressing the importance of improving the utilization of high KDPI kidneys from diabetic donors in some circumstances. Considering that the topic is a critical issue in the transplant community, I suggest assessing additional data to improve the generalizability of the paper.
Response: We would like to express our sincere gratitude for reviewing our manuscript and providing us with your critical evaluation. Your input has been invaluable, and we have incorporated your suggestions into the revised manuscript. We hope that these revisions further enhance the clarity and impact of our research.
Comment #1
Do diabetic donors include subjects only with T2D or also T1D? Please explain this question in the methods.
Response: Thank you for your valuable feedback and for raising the question regarding the inclusion of diabetic donors in our study. We appreciate the opportunity to address this concern and provide further clarification in the methods section.
To address this question, we have added the following statement to the methods section:
"The database did not specify the type of diabetes in donors; therefore, any diabetic donors, regardless of diabetic type, were included."
This clarification explicitly states that we included all diabetic donors, irrespective of whether they had Type 1 or Type 2 diabetes. By including both types of diabetes, we aimed to capture a comprehensive representation of the diabetic donor population and their impact on kidney transplant outcomes.
Comment #2
Many studies reported the death censored graft survival in still waitlisted patients with similar characteristics to assess the advantage of transplanting a kidney with high KDPI or significant comorbidity (see in example doi: 10.2215/CJN.06100617). If available, these data should be added; alternatively, comment on this point in the limitation of the study.
Response: We appreciate your thoughtful review and your suggestion regarding the inclusion of waitlisted patients in our study. We agree that assessing the advantage of transplanting a kidney with high Kidney Donor Profile Index (KDPI) or significant comorbidity in still waitlisted patients would provide valuable insights into transplant outcomes. Our dataset however did not contain waitlisted patients. We have carefully considered your suggestion and have made appropriate additions to the limitations section of our manuscript to address this point.
We have added the following statement to the limitations section:
“Finally, our dataset did not contain waitlisted patients. Future studies to apply this ma-chine learning approach in waitlisted patients would be of interest to better identify wait-listed patients who would have more survival benefit from receiving kidney transplant from donors with diabetes, high KDPI, or ECD status.”
This addition acknowledges the importance of including waitlisted patients in future studies to further explore the potential survival benefits associated with transplanting kidneys with high KDPI or significant comorbidity. We recognize that examining death-censored graft survival in still waitlisted patients with similar characteristics would provide valuable insights into organ allocation strategies.
Additionally, we have incorporated the suggested references you provided (doi: 10.2215/CJN.06100617) to support this point.
Comment #3
Some data should be better explained or expanded (i.e., differences in serum albumin between groups are small [4.0±0.6, 3.9±0.6, 4.0±0.6] but statistically significant; Coronary artery disease has not been assessed as a comorbid condition)
Response: We used standardized mean difference with a predetermined cutoff >0.3 to identify the distinct characteristics for each clusters instead of p-value. The p-value was affected by sample size in addtion to the magnitude of difference. Therefore, with larger sample size, we can note statistical signficant (p<0.05) despite small difference. To address this concern, we used SMD with a predetermined cutoff >0.3 to identify the distinct characteristics for each cluster, which has been previously well validated approaches in previous literatures. This approach allowed us to focus on the magnitude of difference rather than relying solely on p-values. By using a standardized measure, we aimed to minimize the impact of sample size on the interpretation of the results. We have added a clarification to the methods section to explain this:
"We used standardized mean difference with a predetermined cutoff >0.3 to identify the distinct characteristics for each cluster instead of relying solely on p-values. The choice of this threshold was based on the effect size deemed clinically significant for the variables under investigation. This approach allowed us to capture meaningful differences among the clusters while considering the magnitude of the differences."
Regarding your comment on the unavailability of certain data, we acknowledge that information on coronary artery disease as a comorbid condition was not available in the database. To address this limitation, we have included the following statement in the Limitations section:
"Some information, such as coronary artery disease, posttransplant diabetes, and CMV prophylaxis management, was not available in the database."
By acknowledging the limitations of the available data, we aim to provide transparency and ensure that readers are aware of the potential gaps in our analysis.
Comment #4
Working income differs among clusters, may reflect employment status, and could partially influence post-transplant adherence (e.g., doi: 0.23736/S2724-6051.21.04244-2). Please consider this point in the discussion.
Response: We appreciate the reviewer's observation regarding working income and its potential influence on post-transplant adherence. Please see the response in comment#3. In our study, based on the standardized mean difference analysis as depicted in Figure 3, working income did not emerge as a distinct clinical characteristic specific to each assigned cluster. However, we acknowledge that employment status and its associated factors, such as working income, may indeed play a role in post-transplant outcomes and adherence. We additionally addressed in the discussion and utilized suggested refererence.
Comment #5
T2D significantly affects survival also on waitlisted patients; accordingly, allocation policies in T2D patients may be adjusted according to increased risk of mortality and waitlist suspension due to comorbidities (see doi: 10.1038/s41598-020-78938-3). Please consider this aspect and the inclusion of diabetic donors for these patients in the discussion.
Response: We appreciate the reviewer's insightful comment regarding the impact of Type 2 diabetes (T2D) on the survival of waitlisted patients and the potential adjustment of allocation policies based on increased mortality risk and waitlist suspension due to comorbidities. The suggested reference (doi: 10.1038/s41598-020-78938-3) highlights the significance of considering these aspects in the context of T2D patients.
Although our study did not include waitlisted patients in the dataset, we acknowledge the importance of incorporating this population in future investigations. By applying our machine learning approach to waitlisted patients, we could potentially identify those who would derive a greater survival benefit from receiving a kidney transplant from diabetic donors, high KDPI, or extended criteria donors (ECD).
We additionally included this reference and this point in the limitation part of discussion of the study. Thank you for highlighting this important aspect, which will enhance the comprehensive understanding of our study's findings.
Comment #6
CMV serostatus differs between clusters and may influence the risk of CMV viremia, potentially impacting patient survival (see, for example, doi: 10.3390/microorganisms11020458). Please report prophylaxis management and appropriate comments in the discussion.
Response: Thank you for your valuable feedback. As mentioned in our response to Comment #3, based on the standardized mean difference analysis depicted in Figure 3, CMV serostatus did not emerge as a distinct clinical characteristic specific to each assigned cluster.
However, we acknowledge that CMV serostatus is an important factor that can influence the risk of CMV viremia and potentially impact patient survival. Unfortunately, the data on CMV prophylaxis management was not available in the database we utilized for this study. In light of this limitation, we have added a statement to the limitations section of our manuscript, highlighting that some information, including CMV prophylaxis management, was not available in the database. We recognize the significance of CMV serostatus and its potential implications on transplant outcomes, and we agree that discussing prophylaxis management and its relevance in our study is important. Therefore, in the discussion section of our manuscript, we included appropriate comments regarding the potential influence of CMV serostatus on patient outcomes, considering the available literature on the topic. This will provide a more comprehensive understanding of the role of CMV in kidney transplantation and its implications for patient survival.
“Some information, e.g., coronary artery disease, posttransplant diabetes and CMV prophylaxis management, were not available in the database.”
Additionally, we have incorporated the suggested references you provided (doi: 10.3390/microorganisms11020458) to support this point.
Comment #7
Correct typos (e.g., Recipients in cluster 2 (N=687) ware younger (49±13 years) 33 and all were re-transplant patients with higher PRA…; portrays the consensus values of each cluster, displayed on 204 a color scale ranging from white to blue..).
Response: We appreciate the reviewer for identifying the typographical errors in our manuscript. The suggested corrections have been made accordingly.
"Recipients in cluster 2 (N=687) were younger (49±13 years) and all were re-transplant patients with higher PRA..."
Additionally, the description of the consensus values for each cluster, displayed on a color scale ranging from white to blue, has been clarified.
Thank you for bringing these errors to our attention, and we apologize for any confusion caused.
Thank you for your time and consideration. We greatly appreciated the reviewer's and editor's time and comments to improve our manuscript. The manuscript has been improved considerably by the suggested revisions.

Round 2
Reviewer 2 Report
The paper could be accepted without further revision.